# Evolutionary Algorithm-Based Design and Performance Evaluation of Wood–Plastic Composite Roof Panels for Low-Cost Housing

**DOI:** 10.3390/polym17060795

**Published:** 2025-03-17

**Authors:** Bassel Abdelshahid, Khaled Nassar, Passant Youssef, Ezzeldin Sayed-Ahmed, Mohamed Darwish

**Affiliations:** 1Construction Engineering Department, American University in Cairo, New Cairo 11835, Egypt; bassel.harby@aucegypt.edu (B.A.); knassar@aucegypt.edu (K.N.); eysahmed@aucegypt.edu (E.S.-A.); 2Civil Engineering Department, German University in Cairo, New Cairo 11835, Egypt; passant.ahmed@guc.edu.eg

**Keywords:** evolutionary algorithms, low-cost housing, structural optimization, wood–plastic composites

## Abstract

Wood–plastic composites (WPCs) have emerged as a sustainable and cost-effective material for construction, particularly in low-cost housing solutions. However, designing WPC panels that meet structural, serviceability, and manufacturing constraints remains a challenge. This study focused on optimizing the cross-sectional shape of WPC roof panels using evolutionary algorithms to minimize material usage while ensuring compliance with deflection and stress constraints. Two evolutionary algorithms—the genetic algorithm (GA) and particle swarm optimization (PSO)—were employed to optimize sinusoidal and trapezoidal panel profiles. The optimization framework integrated finite element analysis (FEA) to evaluate structural performance under uniformly distributed loads and self-weight. The modulus of elasticity of the WPC material was determined experimentally through three-point bending tests, ensuring accurate material representation in the simulations. The trapezoidal profile proved to be the most optimal, exhibiting superior deflection performance compared with the sinusoidal profile. A comparative analysis of GA and PSO revealed that PSO outperformed GA in both solution optimality and convergence speed, demonstrating its superior efficiency in navigating the design space and identifying high-performance solutions. The findings highlight the potential of WPCs in low-cost housing applications and offer insights into the selection of optimization algorithms for similar engineering design problems.

## 1. Introduction

Wood–plastic composites (WPCs) have emerged as a promising sustainable material, combining recycled wood fibers and thermoplastic polymers to create environmentally friendly and cost-effective alternatives to traditional construction materials [1]. With growing global emphasis on reducing carbon footprints and promoting circular economies, WPCs are increasingly recognized for their potential in structural applications [2]. Their advantages—including resistance to moisture [3], weathering [4,5], and pests [6,7], coupled with low maintenance requirements—make them ideal for load-bearing structures such as roofing panels, where durability and affordability are critical.

To understand and maximize their behavior for structural uses, the mechanical performance of wood–plastic composites (WPCs) has been thoroughly investigated [8,9]. Key mechanical properties—such as the modulus of elasticity, ultimate strength, and fracture toughness—have been shown in experimental studies using tensile, flexural, and impact testing to be greatly influenced by factors including fiber orientation, interfacial bonding quality, and processing conditions during manufacture [10,11,12,13]. Under constant load, the viscoelastic character of the polymer matrix, together with the hygroscopic behavior of the wood fibers, causes time-dependent deformation events, including creep and fatigue [14,15,16,17]. The long-term structural performance of WPCs depends critically on these effects. Together, these material properties determine not only how much weight a WPC-based part can hold at first, but also how long it will last and how well it can be used in different environments. Apart from experimental studies, sophisticated numerical techniques—such as finite element analysis (FEA)—have been applied to replicate stress distribution and deformation mechanisms under challenging loading conditions [18,19]. These simulations make WPCs’ design and dependability in structural uses better by giving detailed information on how to improve performance and predict failure. The integration of experimental and computational studies has deepened the knowledge of WPCs’ behavior, thus enabling their acceptance in many structural systems. Where the twin concerns of sustainability and cost efficiency are especially important, applications include roofing panels [20], shear walls [21,22], and columns [23,24]. A recent study by Darwish et al. (2025) [25] found that using WPCs for the roof with concrete masonry block walls reduced costs to 7731 EGP/m^2^, compared with 8017 EGP/m^2^ for a conventional reinforced concrete skeleton. The highest cost, 11,574 EGP/m^2^, occurred when WPCs were used for both walls and roof.

Structural optimization problems can be broadly categorized into size, shape, and topology optimization [26], each focusing on distinct aspects of a design. Size optimization involves adjusting critical dimensions—such as cross-sectional areas—to ensure that the components meet the strength and deflection requirements, whereas shape optimization refines the external contours to enhance stress distribution and overall performance [27]. Topology optimization, on the other hand, seeks the optimal material distribution within a prescribed design space, often yielding innovative and non-intuitive configurations that maximize efficiency under complex loading scenarios [28]. To solve these problems, two principal classes of methods are employed: mathematical optimization techniques and metaheuristic optimization techniques [29,30]. Mathematical optimization methods, which include gradient-based algorithms, linear and non-linear programming, and sequential quadratic programming, are capable of rigorously identifying optimal solutions when the objective functions and constraints are smooth and differentiable. However, many structural optimization problems involve non-convex search spaces, discrete design variables, or non-smooth, non-differentiable functions, which limit the effectiveness of these classical methods. In these challenging cases, metaheuristic techniques such as genetic algorithms and particle swarm optimization become valuable tools [31,32,33,34,35]. Although they do not guarantee a global optimum, they efficiently explore complex, multimodal design spaces and are well-suited to finding high-quality near-optimal solutions where classical methods struggle. Constraints in these optimization frameworks typically encompass mechanical performance limits like stress and deflection thresholds, buckling resistance, fatigue life, as well as manufacturing and cost considerations, ensuring that the optimized designs are both efficient and practical [36,37,38,39,40].

Despite the increasing adoption of wood–plastic composites (WPCs) in sustainable construction, the application of optimization techniques for enhancing their load-bearing performance remains relatively underexplored. While some research has addressed the optimization of WPCs’ mechanical properties [41,42,43,44,45], only a limited number of studies have systematically employed structural optimization methods to refine the geometric design parameters for load-bearing applications. Zhang et al. [46] investigated the structural design of wood–plastic composite floors by analyzing geometric parameters such as size, shape, and cavity patterns, and found that selecting a design with the largest total cavity area via a method of exhaustive search can optimize strength while saving up to 22% of the material compared with a solid cross-section. Similarly, Shanmugasundaram et al. [47] conducted a parametric study using numerical experiments in LS-DYNA to optimize the cross-sectional area and span of WPC decking structures, achieving a hollow elliptical channel design that reduced the weight by 45% compared with a solid cross-section while meeting the allowable deflection and stress criteria for outdoor applications. Soury et al. [48] developed a multi-objective micro-genetic algorithm integrated with FEA to determine a profile that minimizes mass and deflection while maximizing load capacity, resulting in a pallet weighing less than 20 kg with bending and distributed loading strengths exceeding 500 kg and 2000 kg, respectively. Despite these promising advancements in optimizing WPC components for various load-bearing applications, to the best of the authors’ knowledge, no study has yet considered the structural optimization of wood–plastic composite roof panels’ cross-sectional profiles for low-cost housing while concurrently addressing manufacturing and serviceability constraints.

To address these gaps, this study utilized the genetic algorithm (GA) and particle swarm optimization (PSO) to optimize the cross-sectional profiles of wood–plastic composite (WPC) roof panels for low-cost housing applications. The optimization framework incorporates deflection constraints with key manufacturing parameters—such as wall thickness and extrusion die dimensions—to ensure that the designs are both structurally robust and production-feasible. Finite element analysis (FEA) was utilized to accurately calculate the deflections and stress distributions under realistic loading conditions. The flexural properties of the extruded WPC panels, determined through three-point bending tests, are were as key material inputs in the finite element analysis, ensuring accurate simulation of the panel behavior under load. Three distinct profile geometries—sinusoidal, trapezoidal, and triangular—were investigated, with decision variables defining their respective shape parameters. The key findings demonstrate the superiority of trapezoidal profiles in balancing stiffness and material efficiency, while a comparative analysis of GA and PSO reveals trade-offs between computational speed and solution robustness.

## 2. Materials and Experimental Methods

### 2.1. Material Preparation and Manufacturing Process

The wood–plastic composite (WPC) was formulated with 60 wt.% high-density polyethylene (HDPE) and 40 wt.% sawdust. The raw materials were meticulously combined and processed using a co-rotating twin-screw extruder with a length-to-diameter (L/D) ratio of 40:1 to ensure homogeneous mixing and uniform dispersion of the polymer and lignocellulosic components. A controlled temperature range of 165 °C to 185 °C was used throughout the barrel zones of the extrusion process to improve the melting properties of HDPE and slow down the thermal breakdown of the sawdust. A screw speed of 50 rpm was sustained to guarantee uniform material flow and sufficient shear mixing. The extruded profiles were cooled in room air to keep their shape after they came out of the die. They were then cut to standard specimen sizes according to the applicable mechanical testing protocols so that they could be studied further.

### 2.2. Three-Point Bending Test

Mechanical testing was performed in compliance with ASTM D1037 [49] without any deviations utilizing a universal testing machine (Model 810) manufactured by MTS Systems Corporation located in 14000 Technology Drive Eden Prairie, MN 55344. The load head was operated at a constant crosshead velocity of 6 mm/min to guarantee consistent application of the bending force. Specimens were constructed to the standard dimensions outlined by ASTM D1037, and the experimental arrangement is illustrated in Figure 1. During the examination, real-time load and deflection metrics were documented for future analysis. We calculated the modulus of elasticity (E) in bending using classical beam theory, as shown by the following equation(1)$$\delta =\frac{PL^{3}}{48EI}$$
where δ is the maximum midspan deflection, P is the maximum applied load, L is the span, and I is the second moment of area about the neutral axis.

## 3. Optimization Algorithms and Problem Formulation

The optimization of the wood–plastic composite (WPC) roof panel was carried out using two population-based metaheuristic algorithms: the genetic algorithm (GA) and particle swarm optimization (PSO). Both algorithms were implemented using the Pymoo library [50], a Python-based framework for multi-objective optimization. The theoretical foundations, parameter settings, and mathematical formulations of these algorithms are described below.

### 3.1. Genetic Algorithm (GA)

The genetic algorithm (GA) is inspired by the principles of natural selection and evolution [51]. It operates on a population of candidate solutions, each represented as a set of design variables. The algorithm iteratively evolves the population over successive generations using three primary operators: selection, crossover, and mutation. During the selection phase, individuals with higher fitness values (i.e., those that better satisfy the objective and constraints) are more likely to be chosen as parents for the next generation. The crossover operator, typically with a probability ranging from 0.7 to 1.0, combines the design variables of two parents to create offspring, introducing diversity into the population. The mutation operator, usually applied with a probability between 0.01 and 0.2, introduces random changes to the design variables, allowing the algorithm to explore new regions of the design space and preventing premature convergence [52]. The fitness of each individual is evaluated on the basis of the objective function and the degree to which it satisfies the constraints. The parameters used for the GA include a population size of 100, a crossover probability of 0.9, and a mutation probability of 0.1. These parameters ensure a balance between exploration (searching new regions of the design space) and exploitation (refining existing solutions) [53].

### 3.2. Particle Swarm Optimization (PSO)

The particle swarm optimization (PSO) algorithm is inspired by the social behavior of birds flocking or fish schooling [54]. It operates on a population of particles, each representing a candidate solution in the design space. Each particle has a position $x_{i}$ and a velocity $v_{i}$, which are updated iteratively based on the particle’s own experience and the experience of the swarm. The position update is governed by the following equations(2)$$v_{i}^{t+1}=w\cdot v_{i}^{t}+c_{1}\cdot r_{1}\cdot \left(p_{i}-x_{i}^{t}\right)+c_{2}\cdot r_{2}\cdot \left(g-x_{i}^{t}\right)$$(3)$$x_{i}^{t+1}=x_{i}^{t}+v_{i}^{t+1}$$
where w is the inertia weight, $c_{1}$ and $c_{2}$ are the cognitive and social acceleration coefficients, $r_{1}$ and $r_{2}$ are random numbers between 0 and 1, $p_{i}$ is the best position found by the particle, and g is the best position found by the swarm. The parameters used for PSO include a population size of 100, and acceleration coefficients $c_{1}$ and $c_{2}$ of 2, as originally used by Kennedy and Eberhart [35] in their initial implementation of the algorithm. Eberhart [55] later identified the optimal range for the inertia weight to be between 0.9 and 1.2, with the value chosen in this paper set to 0.9. These parameters ensure that the algorithm effectively balances exploration and exploitation, allowing it to converge toward an optimal or near-optimal solution [33].

### 3.3. Design Problem Formulation

In this study, a predefined set of cross-sectional shapes—including sinusoidal, trapezoidal, and triangular profiles—was explored to identify the optimal design for WPC roof panels. Each profile was parameterized by discrete design variables that control its geometric features. The optimization problem was formulated as a constrained minimization problem, where the general problem formulation can be stated as follows(4)$$Minimize:f_{Area}\left(X,t\right)$$(5)$$Subject\;to:\left\{\begin{array}{c} \delta \left(X,t\right)\leq \;\delta_{max}\\ t_{min}\leq t\leq t_{max}\\ X_{min}\leq X\leq X_{max} \end{array}\right.$$
where the design variables are the set of geometric parameters $X=\{x_{1},\;x_{2},\;\ldots ,\;x_{n}\}$ and a thickness variable t, which collectively describe the cross-sectional shape and dimensions of the panel. The geometric parameters X represent the shape parameters of the predefined profiles, while the thickness variable t determines the uniform thickness of the panel walls. The objective of the optimization is to minimize the cross-sectional area, which is directly proportional to the material usage and weight of the panel. The panel must meet a deflection constrain where $\delta \left(X,t\right)$ is the maximum midspan deflection under the applied loads, and $\delta_{max}$ is the allowable deflection limit. Additionally, the geometric parameters X must lie within specified bounds $X_{min}\leq X\leq X_{max}$, which are determined by the extrusion die’s dimensions and the need to maintain a feasible and manufacturable shape. Similarly, the thickness t must satisfy $t_{min}\leq t\leq t_{max}$ where the minimum thickness is constrained by the extrudability of the WPC material, and the maximum thickness is limited to avoid excessive material usage.

#### 3.3.1. Sinusoidal Wave Panel

The sine wave panel is characterized by a cross-sectional shape defined by a sinusoidal function, which provides a smooth and continuous profile. The shape is generated using the following mathematical formulation(6)$$y\left(x\right)=A\cdot \sin\left(\frac{2\pi x}{P}\right)$$
where y(x) describes the vertical displacement of the sine wave at a horizontal position x, A is the amplitude of the sine wave, and P is the period of the wave. The amplitude A determines the maximum height of the wave from its midline, while the period P controls the horizontal distance between successive peaks of the wave. These two parameters serve as the primary design variables for the sine wave panel. The sine wave shape is particularly advantageous due to its inherent stiffness and ability to distribute stresses evenly, making it a strong candidate for the WPC roof panels. The relationship between the amplitude and period is illustrated in Figure 2a, which shows the geometric configuration of the sine wave panel and highlights the key variables A and P. The smooth curvature of the sine wave also facilitates the extrusion process, as it avoids sharp corners or discontinuities that could complicate manufacturing.

#### 3.3.2. Triangular and Trapezoidal Panels

The triangular and trapezoidal panels were generated using a modified sawtooth function, which allowed for the creation of both shapes by adjusting the parameters of the function. For a given longitudinal position x, the height y(x) of the initial profile is determined by the sawtooth function modulated by the geometric parameters. The base mathematical formulation for generating the profile is given by(7)yx=tan⁡φ·P·S2πxP4
where the sawtooth function S(θ) with 50% duty cycle is defined as:(8)$$S\left(\theta \right)=\left\{\begin{array}{c} \frac{2\theta }{\pi }\;\;\;\;\;\;\;\;\;\;\;\;for-\pi \leq \theta <0\\ 2-\frac{2\theta }{\pi }\;\;\;\;\;\;\;\;for\;0\leq \theta <\pi \end{array}\right.$$
where φ is the angle of the triangular peaks, and P is the period of the sawtooth function. The sawtooth function generates a triangular wave by default, with sharp peaks and troughs determined by the angle φ and the period P. The trapezoidal shape is derived from the triangular wave by introducing an additional parameter, the amplitude A, which limits the maximum and minimum heights of the wave. The key design variables for these shapes are the angle φ the period P, and the amplitude A, which control the steepness, spacing, and height of the peaks, respectively. The triangular panel is characterized by sharp, linear slopes, while the trapezoidal panel features flat sections at the peaks and troughs, providing additional stiffness and stress distribution. Both shapes are illustrated in Figure 2b,c, which highlights the geometric parameters and the transition from a triangular to a trapezoidal profile.

The design parameters were subject to specific limits to ensure both manufacturability and performance. The maximum allowable deflection was set to span/360 to meet serviceability requirements, in accordance with the Egyptian Code of Practice (ECP 203:2020) for an immediate deflection limit due to live loads [56]. Additionally, the panels were assembled from smaller parts, extruded through a die with fixed dimensions of 300 × 200 mm, and the shape parameters of the profiles are inherently constrained by these extrusion dimensions. A minimum wall thickness of 6 mm was also imposed to ensure the extrudability of the WPC material, as thinner walls may lead to manufacturing defects such as warping or incomplete filling of the die. Table 1 below presents the upper and lower bounds on the decision variables governing the profile geometries.

### 3.4. Finite Element Analysis (FEA) of the Panels

The structural performance of the generated panels was evaluated using Finite Element Analysis (FEA), which provides a detailed assessment of the stress distribution and deflection under applied loads. The modeling and analysis were conducted using FreeCAD version 1.0 [57] as the pre- and post-processing software, with CalculiX version 2.22 [58] serving as the finite element solver. The panels were modeled as simply supported one-way shells, utilizing S8 second-order 8-node shell elements for discretization. These elements were chosen for their ability to accurately capture bending and membrane behavior, which are critical for the analysis of thin-walled structures. The simply supported boundary conditions were modeled by applying a pinned support to the troughs of the wave on one side and a roller support on the other side, ensuring that the panel was free to rotate but restrained from vertical displacement at the supports. A uniformly distributed load of 0.6 kN/m^2^, representing the live load, as per the Egyptian Code of Practice (ECP 201:2012) [59] requirements, was applied to the surface of the panel in the gravity direction. Additionally, the self-weight of the panel was included in the analysis to account for the gravitational effects of the WPC material. These loading and boundary conditions are illustrated in Figure 3, which provides a visual representation of the setup, including the applied load and support configurations.

The material properties utilized in the finite element analysis (FEA) were obtained from the three-point bending test performed on the WPC material. The material was characterized as an isotropic, linear elastic substance to streamline the analysis while ensuring adequate precision for the design optimization procedure. The modeling and analysis workflow was automated using Python version 3.13.2 scripting via the FreeCAD API to improve computational efficiency. This automation enabled the swift creation of finite element models, the application of loads and boundary conditions, and the extraction of essential results, such as maximum deflection and stress. A linear static analysis was conducted to assess the structural response of the panels, focusing on the maximum midspan deflection and stress levels. The researchers used the FEA results to make sure that the deflection limits were followed and that the stress levels stayed within the WPC material’s acceptable range.

### 3.5. WPC Panels’ Optimization Framework

The optimization framework for the wood–plastic composite (WPC) roof panels integrates finite element analysis (FEA) and optimization algorithms into a cohesive workflow to identify the optimal cross-sectional shape and thickness. The process begins with the initialization of design variables, such as amplitude, period, and thickness, which define the geometric parameters of the panel. These variables are used to generate candidate shapes, including sinusoidal, triangular, and trapezoidal profiles, which are then evaluated using FEA to compute their structural performance under applied loads. The FEA results, including maximum deflection and stress, are used to assess compliance with the deflection and stress constraints, as well as manufacturing limits such as minimum wall thickness and the extrusion die’s dimensions. The fitness of each candidate solution is calculated on the basis of the cross-sectional area and the degree of constraint violation, with penalty terms applied to infeasible designs. The optimization algorithms, either the genetic algorithm (GA) or particle swarm optimization (PSO), iteratively update the population of candidate solutions, refining the designs over successive generations until the convergence criteria are met. Algorithm 1 below provides a high-level overview of this process, illustrating the sequence of steps and the interactions among shape generation, FEA, and optimization.
**Algorithm 1.** WPC Roof Panel Optimization
**Input:**

*Panel dimensions**Applied loads**Material properties**Constraints*
**Output:**

*Optimal shape parameters*1**Initialization:***Generate* the *initial set of random shape parameters S*2**while** (*Maximum generations not met*)3
**for** *each parameter set x in S* **do**4

*Generate panel using shape parameters x*5

*Apply loads and boundary conditions*6

*Perform finite element analysis*7

*Evaluate objective function and apply penalties for constraint violation*8
**end for**9
*Generate new solutions set and update S*10**end while**11**return** *Optimal set of shape parameters*

## 4. Results

### 4.1. Experimental Results

The three-point bending tests demonstrated uniform mechanical performance among all the evaluated specimens. The average modulus of rupture (MOR) was calculated to be 42.7 MPa, with a standard deviation of 1.5 MPa, signifying low variability among the samples. The modulus of elasticity (MOE) was recorded at an average of 2.46 GPa, with a standard deviation of 80 MPa, indicating consistent elastic behavior under flexural loading. Additionally, the density of the extruded wood–plastic composite (WPC) panels was roughly 540 kg/m^3^, indicating a consistent material composition. The load–deflection curves illustrated in Figure 4 clearly depict the characteristic bending behavior of the WPC panels, exhibiting a linear elastic region that gradually transitions to failure. The results highlight the reproducibility and mechanical integrity of the extruded WPC panels under flexural stress conditions.

### 4.2. Computational Design and Optimization Results

The optimization process yielded optimal cross-sectional shapes for both trapezoidal and sinusoidal panels, each defined by a unique set of geometric parameters. The flexural properties obtained from the three-point bending tests were incorporated into the finite element models (FEM) to accurately represent the material’s behavior. To enhance computational efficiency and reduce running time, the panel width was initially fixed at 400 mm during the optimization phase. Furthermore, full-scale FEM simulations were conducted on the optimal designs using final panel widths of 1200 mm and 3000 mm to verify the performance and structural integrity of the selected profiles. The optimal parameters for each shape, including amplitude, period, and thickness, are summarized in Table 2, which also provides the corresponding cross-sectional area and moment of inertia.

Both designs met the maximum midspan deflection of 8.33 mm, with the trapezoidal panel exhibiting a maximum deflection of 8.2 and the sinusoidal panel showing a maximum deflection of 7.2 mm. The trapezoidal panel’s angular profile, characterized by flat sections and sharp transitions, provided lower cross-sectional area while abiding with the maximum deflection constraint. In contrast, the sinusoidal panel’s smooth, continuous curvature allowed for a more uniform stress distribution, as discussed in the following section. The cross-sectional shapes of the optimal designs are illustrated in Figure 5, with Figure 5a showing the sinusoidal panel and Figure 5b depicting the trapezoidal panel.

## 5. Discussion

### 5.1. Comparative Structural Analysis of Sinusoidal and Trapezoidal Panels

Finite element analysis (FEA) was conducted to evaluate the stress distribution in the optimized cross-sectional profiles. For both the major and minor principal stresses, the trapezoidal panels exhibit pronounced stress concentrations at the peaks and troughs, with higher peak values observed in the flat regions at the peaks. In contrast, the sinusoidal panel demonstrates smoother stress transitions along its continuous curvature, leading to a more uniform distribution. At midspan, the sinusoidal panel recorded a major principal stress of 2.46 MPa and a minor principal stress of 2.64 MPa, whereas the trapezoidal panel showed higher stress levels, with a major principal stress of 2.73 MPa and a minor principal stress of 2.82 MPa. Figure 6 and Figure 7 present the distributions of the major and minor principal stresses, respectively, for both the sinusoidal and trapezoidal panels.

### 5.2. Algorithmic Convergence and Solution Quality Assessment

A comparative analysis was conducted to evaluate the performance of the genetic algorithm (GA) and particle swarm optimization (PSO) in optimizing the cross-sectional parameters of the WPC panels. For the sinusoidal panel, the GA converged to a solution characterized by a lower panel thickness and a geometry where the amplitude was less than the period. In contrast, PSO identified a solution with a higher panel thickness; while both the amplitude and period were lower than those in the GA solution, the amplitude exceeded the period. For the trapezoidal panel, both algorithms favored designs with lower amplitude values relative to the period; however, the GA solution exhibited a higher thickness, lower amplitudes and periods, and a higher corrugation angle than the PSO solution. While the difference in the objective function values was modest for the sinusoidal panel, it was more pronounced in the trapezoidal configuration—underscoring the superior exploration capability of PSO in navigating the solution space and avoiding local optima. Figure 8 illustrates the evolution of decision variables and the corresponding objective function values over 30 generations for both panel types, with subfigure (a) representing the sinusoidal panel and subfigure (b) representing the trapezoidal panel. Each connected line in the figure represents the values of the decision variables for the best individual in the population at each generation, along with its corresponding deflection and objective function values. The optimal solution found across all 30 generations is highlighted in bold, representing the best-performing configuration achieved during the optimization process.

Figure 9a–d illustrate the convergence behavior of both GA and PSO for the sinusoidal and trapezoidal panels. The PSO algorithm exhibited superior convergence efficiency, achieving optimization within six generations for both panel geometries. This rapid convergence can be attributed to PSO’s velocity-based position updating mechanism and social learning components, which facilitate effective exploration of the continuous design space. The algorithm’s inherent ability to maintain population diversity, coupled with its memory retention of global and personal best positions, enables efficient navigation of the solution landscape. In contrast, the GA demonstrated notably slower convergence rates, requiring 12 and 28 generations for the sinusoidal and trapezoidal panels, respectively. Several factors can explain this difference in performance. First, GA’s discrete nature of genetic operators (crossover and mutation) may not be as effective in navigating continuous design spaces compared with PSO’s continuous position updates. Second, the tournament selection process in GA can lead to premature convergence due to reduced population diversity, particularly in complex solution spaces.

The difference in convergence rates between the sinusoidal and trapezoidal panels’ optimizations (12 versus 28 generations for GA) can be also attributed to the inherent characteristics of their respective solution spaces. The sinusoidal panel’s solution space exhibits higher smoothness and continuity due to its gradual geometric variations, resulting in a more well-behaved optimization landscape. Conversely, the trapezoidal panel’s solution space is characterized by sharper transitions and inherent discontinuities, particularly due to the interplay between the φ and amplitude parameters. These parameters can transform the profile between trapezoidal and triangular configurations, each with distinct structural behaviors. This parametric flexibility introduces multiple regime changes in the structural response, resulting in a more complex solution space with numerous local optima and distinct behavioral zones. The abrupt changes in structural characteristics during these geometric transitions pose additional challenges for optimization algorithms, particularly affecting the GA’s ability to efficiently navigate across these regime boundaries.

Figure 9c further highlights the differing convergence behaviors of PSO and GA in the case of the trapezoidal panel, demonstrating how GA exhibits a slower and more erratic convergence pattern due to its difficulty in escaping local optima. Additionally, Figure 8b shows that the optimal solution obtained by GA had a higher thickness value compared with that of PSO, indicating that GA converged to a local optimum and was unable to further explore the solution space effectively. This limitation arises from GA’s relatively lower exploration capabilities compared with PSO, which allows PSO to achieve a more globally optimal solution by better navigating the complex and discontinuous search space.

## 6. Conclusions

This study introduced an evolutionary algorithm-based method for optimizing the cross-sectional profiles of wood–plastic composite (WPC) roof panels designed for affordable housing. The amalgamation of experimental flexural property assessments, finite element analysis, and evolutionary optimization methodologies, such as the genetic algorithm (GA) and particle swarm optimization (PSO), facilitated the design of effective and manufacturable panel profiles. We outline the principal findings and recommendations below.
Three-point bending tests showed that the WPC material, which is made up of 60% HDPE and 40% sawdust, had consistent mechanical properties. Its average modulus of rupture was 42.7 MPa and its average modulus of elasticity was 2.46 GPa.A range of cross-sectional geometries, such as sinusoidal, trapezoidal, and triangular profiles, was assessed. The trapezoidal shape was seen as the best balance between material efficiency and structural rigidity, so it was chosen as the best option for further development.Particle swarm optimization (PSO) exhibited enhanced efficacy relative to the genetic algorithm (GA) by achieving faster convergence and meeting the deflection constraints in fewer iterations. This efficiency diminishes computational time while preserving the accuracy of the optimization process.Finite element analysis demonstrated unique stress distribution characteristics across the profiles. The sinusoidal profile demonstrated more gradual stress transitions, potentially decreasing the risk of failures associated with stress concentration. Conversely, the trapezoidal profile exhibited elevated peak stresses concentrated in flat areas. Still, these stress levels stayed within the WPC material’s acceptable range, which meant that the structure would remain strong under the expected loading conditions.These findings underscore the effectiveness of integrating experimental characterization, computational modeling, and sophisticated optimization algorithms to improve the performance and manufacturability of WPC roof panels. To make the best use of WPC materials in long-lasting, low-cost housing solutions, future research may focus on improving the way the materials are made and looking into how long they last in different environments.

## Figures and Tables

**Figure 1 polymers-17-00795-f001:**
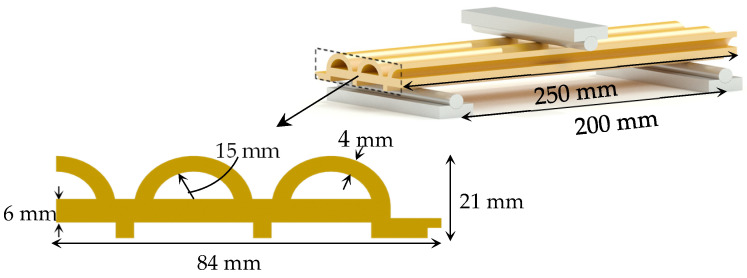
Three-point bending test specimens’ dimensions and experimental setup.

**Figure 2 polymers-17-00795-f002:**
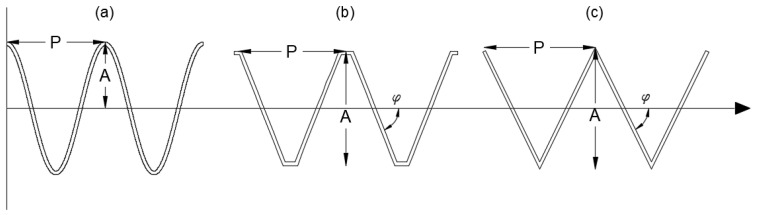
Cross-sectional shape parameter for the sinusoidal (**a**), trapezoidal (**b**), and triangular (**c**) panels.

**Figure 3 polymers-17-00795-f003:**
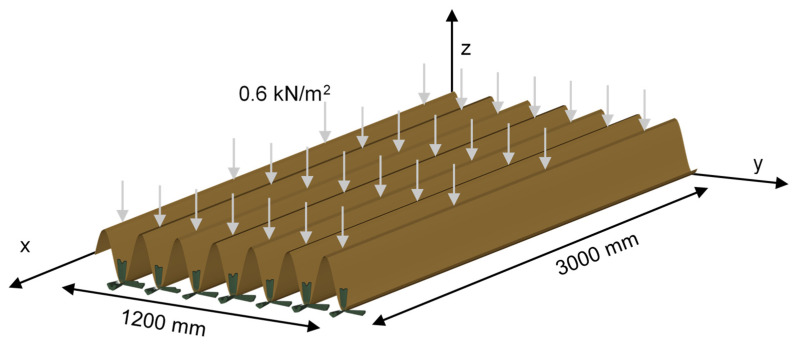
Load configuration and boundary conditions.

**Figure 4 polymers-17-00795-f004:**
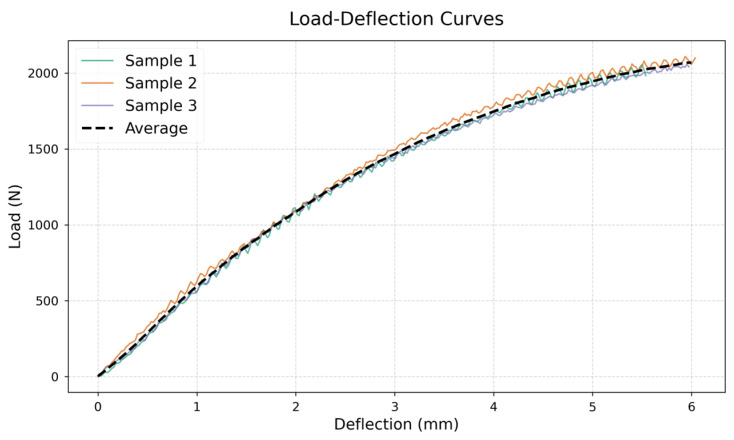
Load–deflection curves for WPC specimens from the three-point bending tests.

**Figure 5 polymers-17-00795-f005:**
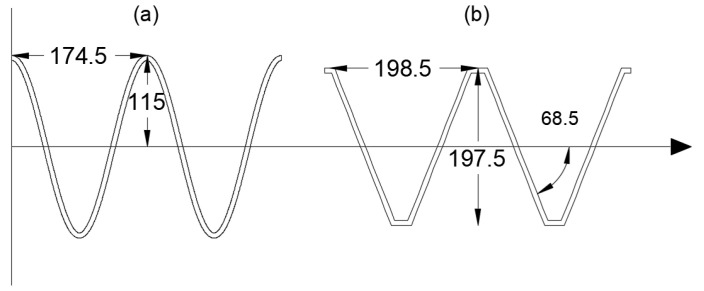
Optimal cross-sectional profiles for the sinusoidal panel (**a**) and the trapezoidal panel (**b**).

**Figure 6 polymers-17-00795-f006:**
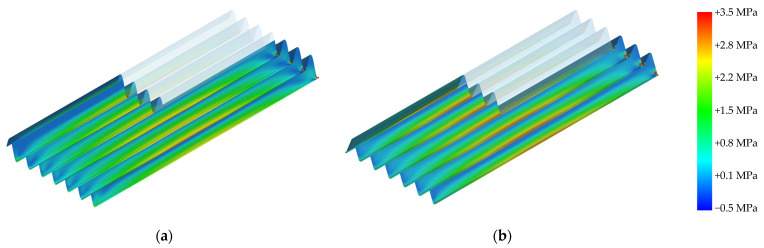
Major principal stress distributions for the sinusoidal (**a**) and trapezoidal (**b**) panels, with midspan stress values of 2.46 MPa (sinusoidal) and 2.73 MPa (trapezoidal).

**Figure 7 polymers-17-00795-f007:**
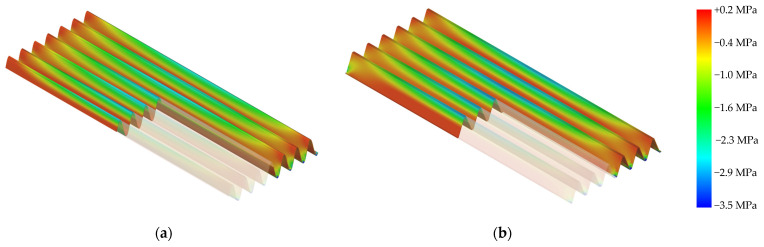
Minor principal stress distributions for the sinusoidal (**a**) and trapezoidal (**b**) panels, with midspan stress values of 2.64 MPa (sinusoidal) and 2.82 MPa (trapezoidal).

**Figure 8 polymers-17-00795-f008:**
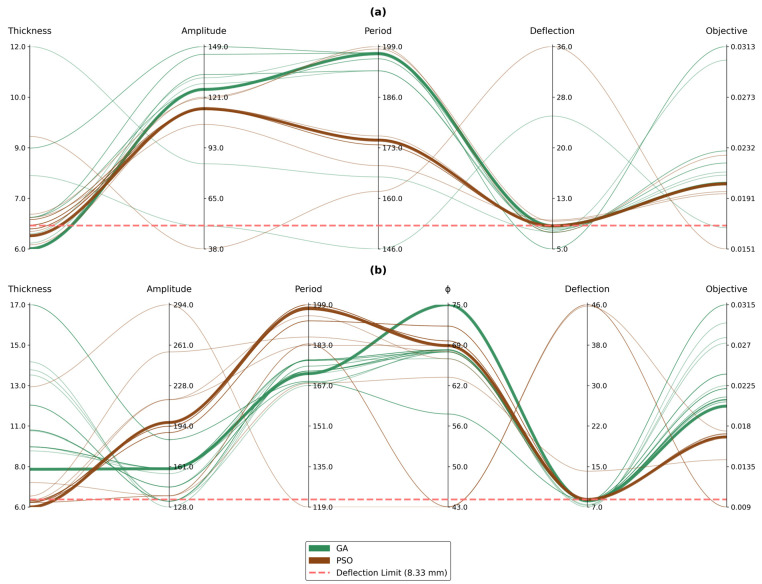
Evolution of the decision variables and objective function values over 30 generations comparing GA’s and PSO’s performance for (**a**) the sinusoidal panel and (**b**) the trapezoidal panel.

**Figure 9 polymers-17-00795-f009:**
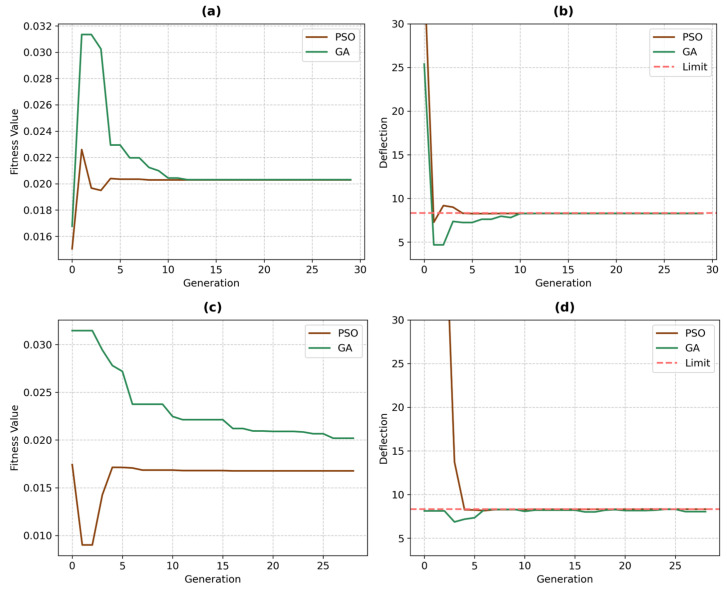
Convergence plots for GA and PSO: (**a**) Objective value vs. generation for the sinusoidal panel, (**b**) maximum deflection vs. generation for the sinusoidal panel, (**c**) objective value vs. generation for the trapezoidal panel, and (**d**) maximum deflection vs. generation for the trapezoidal panel.

**Table 1 polymers-17-00795-t001:** Decision variables’ upper and lower bounds.

Parameter	Lower Bound	Upper Bound
A	20 mm	300 mm
P	50 mm	200 mm
φ	20	85
t	6	50

**Table 2 polymers-17-00795-t002:** Optimal geometric parameters and corresponding structural properties.

Shape	Parameters			Cross-Sectional Properties	
	Amplitude	Period	φ	Area (mm^2^)	$I_{xx}$ (mm^4^)
Sinusoidal	230	174.5	-	22,054	108,328,378
Trapezoidal	197.5	198.5	197.5	17,444	67,553,565

## Data Availability

The raw data supporting the conclusions of this article will be made available by the authors on request.

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
