# Peer review of "Evolutionary Algorithm-Based Design and Performance Evaluation of Wood–Plastic Composite Roof Panels for Low-Cost Housing"

_polymers, 2025, doi:10.3390/polym17060795_

Round 1
Reviewer 1 Report
Comments and Suggestions for Authors
The report presents, for the first time, the optimization of the cross-sectional shape of a WPC panel using GA and PSO algorithms. The findings reveal that PSO is more suitable for optimization in terms of both efficiency and accuracy.
The report is well-structured and clearly presented. However, a few questions need to be addressed before publication:
- Line 59: Including a price comparison with other materials would significantly benefit the readers.
- Line 164: Are there any prior studies that support the choice of GA parameters? Providing references to related research would strengthen the justification.
- Line 176: Please cite a reference or provide reasoning to support the claim that these parameters ensure the algorithm's balancing effectiveness. Additional examples or practical experiences would be valuable to readers.
- Line 250: A citation for FreeCAD is needed.
- Line 258: A citation for ECP is required.
- Line 364: Further explanation of Figure 8 is necessary. How are the optimized variables in the figure connected by the plotted curves?
- Line 382: A more detailed discussion is needed regarding the differing convergence behaviors of PSO and GA in Figure 9(c). What factors contribute to these differences in fitness values?
Author Response
Comment # 1: Line 59: Including a price comparison with other materials would significantly benefit the readers.
Response 1: The authors would like to thank the reviewer for the comment, the cost per square meter was added in lines 60 - 63
Comment # 2: Line 164: Are there any prior studies that support the choice of GA parameters? Providing references to related research would strengthen the justification.
Response 2: The authors would like to thank the reviewer for the comment, a paragraph explaining the range for the parameters is added in lines 163-167 with supporting refences, two additional references were also added to support the claims and choice of parameters [52],[53].
Comment # 3: Line 176: Please cite a reference or provide reasoning to support the claim that these parameters ensure the algorithm's balancing effectiveness. Additional examples or practical experiences would be valuable to readers.
Response 3: The authors would like to thank the reviewer for the comment, a paragraph is added in lines 181-187 to support the choice of the parameters and an additional reference is added [55].
Comment # 4: Line 250: A citation for FreeCAD is needed.
Response 4: The authors would like to thank the reviewer for the comment, citation was added for both FreeCAD and CalculiX, lines 258 and 259
Comment # 5: Line 258: A citation for ECP is required.
Response 5: The authors would like to thank the reviewer for the comment, citation was added in line 267
Comment # 6: Line 364: Further explanation of Figure 8 is necessary. How are the optimized variables in the figure connected by the plotted curves?
Response 6: The authors would like to thank the reviewer for the comment, an explanation for figure 8 was added in lines 272-277
Comment # 7: Line 382: A more detailed discussion is needed regarding the differing convergence behaviors of PSO and GA in Figure 9(c). What factors contribute to these differences in fitness values?
Response 7: The authors would like to thank the reviewer for the comment, a paragraph was added in lines 413-421 linking the discussion in the previous paragraphs to the convergence behaviors of PSO and GA in the case of the trapezoidal panels.

Reviewer 2 Report
Comments and Suggestions for Authors
- The main question addressed by the research.
This study aims to explore the potential of evolutionary algorithms in optimizing the cross-sectional shape of WPC roof panels for low-cost housing. The central research question guiding this study is how to minimize material usage while meeting deflection and stress constraints. To this end, the study compares the performance of GA and PSO in optimizing sinusoidal and trapezoidal panel profiles.
- The topic original or relevant to the field.
The subject matter is pertinent to the field, building on extant research in the area of WPCs and structural optimization. Although WPCs are increasingly employed in construction, the application of optimization techniques to enhance their load-bearing performance in low-cost housing, with a particular focus on cross-sectional shape optimization, is presented as being relatively under-explored. The paper references earlier studies that optimized the mechanical properties of WPCs or specific components, but it emphasizes the innovation of the present study in focusing on the optimization of roof panel cross-sectional profiles while considering manufacturing constraints. Consequently, while the paper may not be entirely original in its constituent parts, the combination and application of these parts to this particular problem within the context of low-cost housing results in a relevant and valuable contribution.
- The subject area compared with other published material.
The present research is situated at the intersection of several subject areas, namely material science (with a particular focus on wood-plastic composites), structural engineering (with regard to structural optimization and finite element analysis), and computer science (with respect to evolutionary algorithms).
- The conclusions consistent with the evidence and arguments presented.
The conclusions arrived at are directly supported by the evidence presented within the research's defined scope. The paper presents a logical progression from experimental data through computational modelling and algorithmic optimization to conclusions that are substantiated by the evidence.
- Appropriate of the references.
It is evident that the references have been selected in a manner that is largely appropriate and relevant to the subject matter. They provide a robust support for the claims that are made throughout the paper and demonstrate a thorough review of the existing literature on WPCs, structural optimization, and evolutionary algorithms.
- Additional remarks, comments.
- In Section 2.2 (Three-point bending test), it is stated that the modulus of elasticity was calculated using classical beam theory. However, it is not clear whether corrections for the effects of transverse shear were taken into account, especially since the WPC material may have a relatively low modulus of elasticity. Furthermore, what assumptions were made in the calculation? How were the measured results processed? Were there any outliers?
- The maximum deflection limits are specified as L/360. The rationality of this choice, given the characteristics of the WPC and the anticipated service conditions, is questionable. Have other criteria, such as tensile and compressive strength, been considered?
- A further issue that must be addressed is the effectiveness of the elemental support paradigm in facilitating the adhesion of realistic roof panels.
Author Response
Comment # 1: In Section 2.2 (Three-point bending test), it is stated that the modulus of elasticity was calculated using classical beam theory. However, it is not clear whether corrections for the effects of transverse shear were taken into account, especially since the WPC material may have a relatively low modulus of elasticity. Furthermore, what assumptions were made in the calculation? How were the measured results processed? Were there any outliers?
Response 1: The authors would like to thank the reviewer for the comment, the test was done according to the ASTM without any deviation from the specified test procedures as specified in lines 137 and 138 that didn’t include any requirements for corrections for the effects of transverse shear. Additionally, the span to depth ratio is significantly large hence negating any probability of shear failure.
Comment # 2: The maximum deflection limits are specified as L/360. The rationality of this choice, given the characteristics of the WPC and the anticipated service conditions, is questionable. Have other criteria, such as tensile and compressive strength, been considered?
Response 2: The authors would like to thank the reviewer for the comment, a reference to the ECP requirements for immediate deflection was added in lines 244-246 in which the L/360 was specified as the serviceability requirement in addition to the strength requirement of satisfactory bending strength compared to the bending stresses.
Comment # 3: A further issue that must be addressed is the effectiveness of the elemental support paradigm in facilitating the adhesion of realistic roof panels.
Response 3: The authors would like to thank the reviewer for the comment. The authors totally agree with the reviewer on the importance of such an issue. This issue together with all other issues related to manufacturing, installation and construction of the proposed system are currently under-study within the current phase of this research project and will be published in a separate publication.
